 **eLIFE**

# Dynamic relocalization of replication origins by Fkh1 requires execution of DDK function and Cdc45 loading at origins

Haiyang Zhang, Meghan V Petrie, Yiwei He, Jared M Peace, Irene E Chiolo, Oscar M Aparicio*

Molecular and Computational Biology Section, Department of Biological Sciences, University of Southern California, Los Angeles, United States

**Abstract** Chromosomal DNA elements are organized into spatial domains within the eukaryotic nucleus. Sites undergoing DNA replication, high-level transcription, and repair of double-strand breaks coalesce into foci, although the significance and mechanisms giving rise to these dynamic structures are poorly understood. In *S. cerevisiae*, replication origins occupy characteristic subnuclear localizations that anticipate their initiation timing during S phase. Here, we link localization of replication origins in G1 phase with Fkh1 activity, which is required for their early replication timing. Using a Fkh1-dependent origin relocalization assay, we determine that execution of Dbf4-dependent kinase function, including Cdc45 loading, results in dynamic relocalization of a replication origin from the nuclear periphery to the interior in G1 phase. Origin mobility increases substantially with Fkh1-driven relocalization. These findings provide novel molecular insight into the mechanisms that govern dynamics and spatial organization of DNA replication origins and possibly other functional DNA elements.

DOI: https://doi.org/10.7554/eLife.45512.001

*For correspondence:
oaparici@usc.edu

## Introduction

The spatial organization of chromosomal DNA elements within the nucleus is thought to derive from and contribute to the regulation of their activity (reviewed in *Shachar and Misteli, 2017*). For example, euchromatin and heterochromatin represent distinct forms of chromatin that are distinguished by their levels of transcriptional activity, replication timing, and subnuclear localization (reviewed in *Caridi et al., 2017*). Chromosomes partition into subdomains ranging from hundreds to thousands of kilobases in length that preferentially self-associate and are consequently referred to as topologically associated domains (TADs) (reviewed in *Zhao et al., 2017*). TAD boundaries correlate closely with replication timing domains, suggesting that replication timing is determined or influenced by this domain structure and/or *vice-versa*.

In budding and fission yeast, specific mechanisms defining replication timing are linked with chromosomal domain organization (reviewed in *Aparicio, 2013*; *Yamazaki et al., 2013*). Rif1, which is highly enriched at telomeres, is globally responsible for delayed replication timing of subtelomeric domains as well as internal late-replicating domains (*Hafner et al., 2018*; *Hayano et al., 2012*; *Peace et al., 2014*; *Tazumi et al., 2012*). Rif1 acts by directly antagonizing replication initiation triggered by Dbf4-dependent kinase (DDK) phosphorylation of MCM helicase proteins (*Davé et al., 2014*; *Hiraga et al., 2014*; *Mattarocci et al., 2014*). Against this inhibitory backdrop, specific origins are selected for early activation by mechanisms involving recruitment of Dbf4 (Dfp1 in fission yeast), which is one of several initiation proteins present in limited abundance and thus rate-limiting for

origin firing (*Mantiero et al., 2011*; *Patel et al., 2008*; *Tanaka et al., 2011*; *Wu and Nurse, 2009*). In *S. pombe*, Dfp1 is recruited to kinetochores through heterochromatin protein Swi6 (*Hayashi et al., 2009*), and in *S. cerevisiae*, kinetochore protein Ctf19 recruits Dbf4 to stimulate firing of origins within ~25 kb of the centromere (*Natsume et al., 2013*), thus ensuring early centromere replication by distinct mechanisms regulating DDK activity. In *S. cerevisiae*, Fkh1 and/or Fkh2 (Fkh1/2) recruits Dbf4 to many origins distributed throughout chromosome arms, thereby ensuring earlier replication of many centromere-distal regions (*Fang et al., 2017*; *Knott et al., 2012*; *Lõoke et al., 2013*; *Ostrow et al., 2014*).

Chromosome conformation capture experiments suggest that early-firing origins cluster spatially in G1-phase prior to initiation and this clustering is dependent on Fkh1/2 (*Duan et al., 2010*; *Knott et al., 2012*). These studies also indicated that early origins generally occupy a distinct space than late origins. Further studies suggest that Fkh1/2 are enriched at TAD boundaries and control contacts among origins within TADs (*Eser et al., 2017*). The distinct spatial distributions suggested by these recent studies are in accord with earlier studies that examined the subnuclear distribution of individual origins by fluorescence microscopy. These seminal studies from Heun and Gasser showed that late-firing origins typically associate with the nuclear periphery during G1 phase whereas early-firing origins typically are found in the nuclear interior during G1 (*Heun et al., 2001a*; *Heun et al., 2001b*). Despite the observed correlations between origin localization in G1 and firing time in S, the main origin timing determinants mentioned above had not been elucidated and have not been examined for their impact on subnuclear localization of replication origin. In this study, we examined how origin stimulation by Fkh1 determines subnuclear origin localization. Our results suggest that origin relocalize from the nuclear periphery upon execution of the DDK-dependent step of replication initiation, which is stimulated by Fkh1. This may represent the initial stages in the coalescence of replication origins into clusters that will become replication factories.

## Results

### Fkh1-induced origin activation re-positions a subtelomeric origin in G1 phase

The association between replication timing and subnuclear localization of replication origins and the requirement of Fkh1/2 for the clustering of early-firing replication origins according to chromosome conformation capture studies led us to examine whether Fkh1 has any role in establishing the spatial positioning of origins within the nucleus. We adapted a system that we recently engineered that restores origin timing by induction of *FKH1* expression in G1-arrested *FKH1/2* mutant cells (*Peace et al., 2016*). This system has Fkh-activated origin *ARS305* moved into a well characterized, late-replicating, subtelomeric region of chromosome V-R, replacing the endogenous, late-firing *ARS501* (*Figure 1A*). In this context, we showed previously that $ARS305^{V-R}$ fails to replicate early in *fkh1Δ fkh2Δ* cells. However, induction of *FKH1* expression in these cells in G1-phase results in early-firing of $ARS305^{V-R}$ in the ensuing S-phase. In the current study, we used *fkh1Δ fkh2-dsm instead of* *fkh1Δ fkh2Δ* cells; *fkh1Δ fkh2-dsm* cells are essentially null for replication timing control, but exhibit more normal growth and particularly, more normal cell and nuclear morphologies favorable for cytological analysis (*Ostrow et al., 2017*). To locate $ARS305^{V-R}$ (or *ARS501*) in vivo, we introduced tandem repeats of tetO binding sites adjacent to the origin and expressed TetR-Tomato protein (*Figure 1A*); we also expressed Nup49-GFP (Nup49 is a nuclear pore protein) to illuminate the nuclear envelope (*Belgareh and Doye, 1997*).

Microscopic examination of cells showed a single Tomato focus per undivided nucleus (*Figure 1B*). Images of cells from an unsynchronized population were sorted according to budding morphology, which is reflective of cell cycle progression. The localization of the $ARS305^{V-R}$-Tomato focus correlates with cell cycle stage, showing primarily peripheral localization in unbudded and small-budded cells and interior localization in larger-budded cells (*Figure 1B*). This is consistent with previous studies showing peripheral localization of subtelomeric/late-firing origins in G1 followed by relocalization to the interior during S phase (*Heun et al., 2001a*). Because origin timing is normally established in G1, we focused further analysis on origins in G1 phase cells.

In G1-arrested *fkh1Δ fkh2-dsm* cells, almost all cells exhibited peripheral localization of $ARS305^{V-R}$ (*Figure 1C*, left panel). Induction of *FKH1*, however, resulted in an increase in the proportion of cells

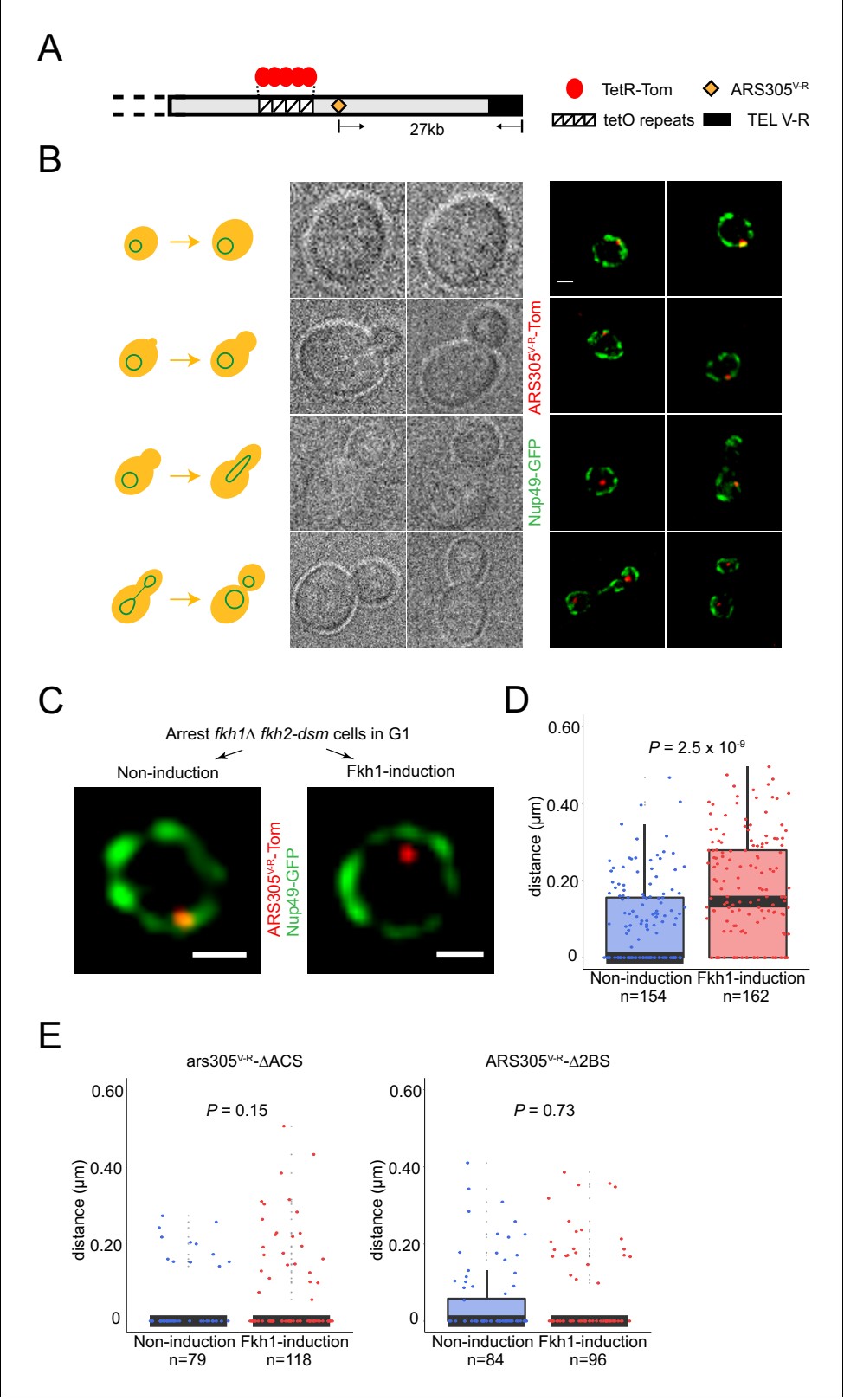

**Figure 1.** Fkh1-induced origin activation re-positions a subtelomeric origin in G1 phase. (**A**) Schematic of chromosome V-R showing tetO repeats inserted adjacent to the *ARS501* locus, which has been replaced with *ARS305* (designated *ARS305^(V-R)*); TetR-Tomato binds to and illuminates the locus as a single focus. (**B**) Images of cells from an unsynchronized culture of strain HYy132 (*fkh1Δ fkh2-dsm ARS305^(V-R)-Tomato NUP49-GFP GAL-FKH1*)
*Figure 1 continued on next page*

*Figure 1 continued*

are shown sorted according to cell morphology; all images are at the same magnification: scale bar = 0.5 µm. (C) *FKH1* induction scheme: HYy132 cells grown at 25°C in raffinose medium were arrested in G1 phase with 1x α-factor 2.5 hr, incubated an additional 2 hr in raffinose (Non-induction) or galactose (Fkh1-induction) with 1.7x α-factor, and images of live cells captured, examples of which are shown; scale bar = 0.5 µm. (D) The shortest distance from the *ARS305$^{V-R}$-Tomato* focus to the nuclear periphery (Nup49-GFP) in each cell was measured and plotted as quartile boxplots (median shown as thick black segment) for non-induction and *FKH1*-induction; the result of a z-test comparing the two distributions is given as P. (E) Cells of *fkh1Δ fkh2-dsm GAL-FKH1 NUP49-GFP* strains HYy119 (*ars305-ΔACS$^{V-R}$-Tomato*) and HYy120 (*ars305-Δ2BS$^{V-R}$-Tomato*) were treated and analyzed as above.

DOI: https://doi.org/10.7554/eLife.45512.002

The following figure supplement is available for figure 1:

**Figure supplement 1.** Fkh1-induction is required to re-position a subtelomeric origin in G1 phase.

DOI: https://doi.org/10.7554/eLife.45512.003

with non-peripheral positioning of *ARS305$^{V-R}$* (*Figure 1C*, right panel), suggesting that origin relocalization is associated with initiation timing re-programming by *FKH1*. We confirmed that relocalization is a direct result of *FKH1* induction by demonstrating that neither the induction scheme (raffinose -> galactose) with a strain lacking inducible *FKH1* nor a non-inducing change to a more favorable carbon source (raffinose -> dextrose) resulted in origin relocalization (*Figure 1—figure supplement 1*). To confirm the change in origin localization resulting from *FKH1* induction, we created three-dimensional image reconstructions from confocal z-stacks and measured the shortest distance in three dimensions from the *ARS305$^{V-R}$* focus to Nup49-GFP signal in the nuclear envelope amongst populations of cells. Statistical analysis of these measurements shows a significant increase in the distances associated with *FKH1*-induction versus non-induction (*Figure 1D*).

We tested whether the function of *ARS305* is required for relocalization by introducing into the V-R locus *ARS305* bearing a mutation of the ARS consensus sequence (ACS) (ars305-ΔACS), which is essential for ORC binding and origin function. Disruption of *ARS305* function not only eliminated its relocalization in response to *FKH1* induction but also resulted in an even more peripheral distribution, suggesting that a functional origin is required for relocalization away from the periphery (*Figure 1E*). We also tested *ARS305* with mutations of two proximal Fkh1/2 binding sites (ars305-Δ2BS), which retains origin function but is delayed in activation at its normal locus (*Knott et al., 2012*); *ars305-Δ2BS$^{V-R}$* did not relocalize upon *FKH1* induction, confirming that Fkh1 acts through direct binding in cis to *ARS305$^{V-R}$* (*Figure 1E*).

In the experiments above, relocalization of *ARS305$^{V-R}$* involved induction of *FKH1* from the *GAL1/10* promoter, which results in higher than normal levels of Fkh1 protein (*Peace et al., 2016*). To determine whether this overabundance of Fkh1 was required for the origin relocalization, we compared localization of *ARS305$^{V-R}$* with *ARS501* in cells with native *FKH1* (and *FKH2*) expression (*Figure 2A*). The analysis showed that *ARS305$^{V-R}$* was significantly more distant from the nuclear periphery than *ARS501* (*Figure 2B*). We also analyzed origin timing of *ARS305$^{V-R}$* in *FKH1* (*fkh2-dsm*) versus *fkh1Δ* (*fkh2-dsm*) cells by quantitative BrdU immunoprecipitation (QBU) of cells released from G1 phase into hydroxyurea (HU), in which early but not late origins fire efficiently. We found that *ARS305$^{V-R}$* fired efficiently in HU in *FKH1* but not *fkh1Δ* cells (*Figure 2C* and *Figure 2—figure supplement 1*). Thus, normal Fkh1 levels are able to overcome the effect that subtelomeric location has on subnuclear localization and initiation timing of *ARS305$^{V-R}$*.

## Fkh1 globally regulates subnuclear positioning of early origins in G1 phase

We tested whether the Fkh1-dependent localization of *ARS305$^{V-R}$* is also responsible for *ARS305* localization when residing at its native locus more distal from the telomere. We inserted a lacO array near *ARS305* and expressed LacI-GFP and Nup49-GFP; imaging showed that the LacI-GFP focus is clearly distinguishable from the more diffuse Nup49-GFP signal (*Figure 3A*). Consistent with previous analysis, *ARS305* was non-peripheral in most G1-arrested *WT* cells (*Figure 3A*) (*Heun et al., 2001a*), however, deletion of *FKH1* significantly increased the proportion of cells in which *ARS305* was closer to the periphery (*Figure 3A*). Consistent with this requirement for *FKH1*, elimination of

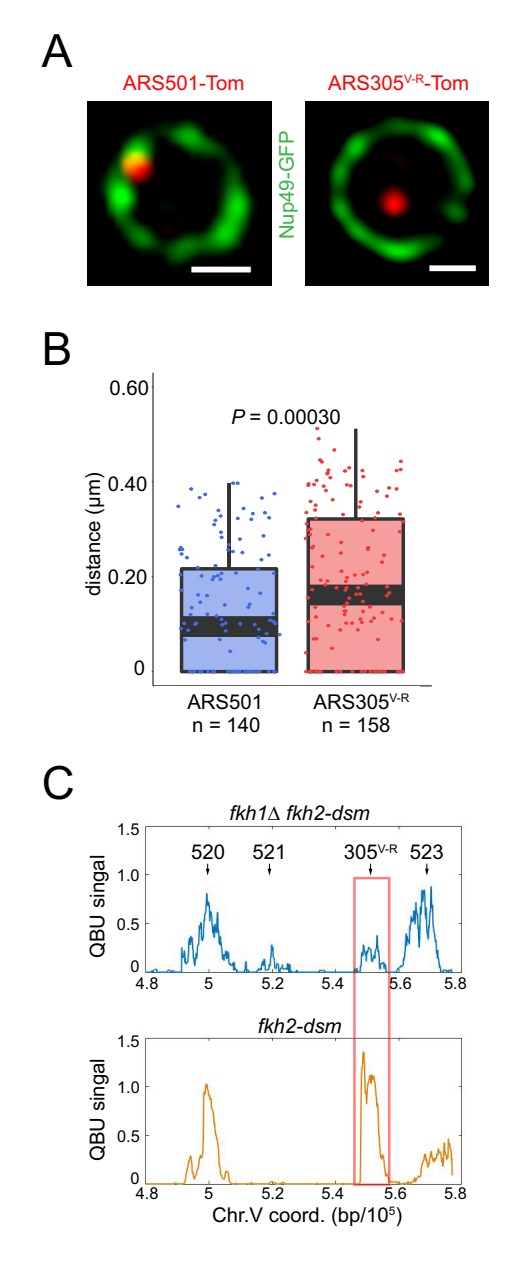

**Figure 2.** Normal dosage of Fkh1 is sufficient to relocalize *ARS305^{V-R}* and advance its firing time. (**A**) HYy160 (*ARS501-Tomato NUP49-GFP*) and HYy157 (*ARS305^{V-R}-Tomato NUP49-GFP*) cells were arrested in G1 phase at 25°C with 1x α-factor 2 hr and images were collected; scale bar = 0.5 μm. (**B**) Distances from origin foci to nuclear periphery were determined, plotted as quartile boxplots, and analyzed by a z-test. (**C**) Quantitative BrdU-IP-Seq (QBU) analysis was performed with *ARS305^{V-R}*-bearing strains HYy113 (*fkh2-dsm*) and HYy38 (*fkh1Δ fkh2-dsm*) after G1 block-and-release into hydroxyurea in the presence of BrdU; averaged data from three experimental replicates was plotted for the V-R region with the positions of several

the Fkh1/2 binding sites in *ARS305* also resulted in peripheral localization (*Figure 3—figure supplement 1A*). Moreover, *ars305-ΔACS*, which lacks origin function (and Fkh1/2 binding) exhibits an even more peripheral distribution (*Figure 3—figure supplement 1A*), suggesting that origin function is required for interior localization in G1 and that Fkh1/2 stimulates this localization. Previous analysis showed that *ARS305* initiation timing was significantly delayed in the absence of *FKH1* or Fkh1/2 binding sites (*Knott et al., 2012*), so once again we observe a *FKH1*-dependent relationship between subnuclear localization in G1 phase with replication initiation timing in S phase.

To determine whether other Fkh-activated origins' localizations are also determined by *FKH1*, we performed similar tests by inserting a tetO array adjacent to a few additional Fkh-activated origins, expressing TetR-Tomato and Nup49-GFP, and deleting *FKH1*. Like *ARS305*, *ARS1303* and *ARS1103* were located closer to the periphery in G1-arrested *fkh1Δ* versus *WT* cells (*Figure 3B*). *ARS305*, *ARS1303* and *ARS1103* are relatively telomere proximal, residing 39, 32, and 56 kb from the nearest telomere respectively (the lacO or tetO arrays add ~14 or 16 kb, respectively to these distances), which might constrain the extent to which *FKH1* influences their positioning. To address this possibility, we tested localization of several additional Fkh-activated origins that are more distal from telomeres, including: *ARS710*, *ARS718*, and *ARS1018*, residing at 204, 421, and 205 kb from the nearest telomere. All of these origins show significant reduction in distance from the nuclear periphery upon deletion of *FKH1* (*Figure 3C*). We also observed similar results with *ARS305* and *ARS710* in G1 cells from an unsynchronized population (*Figure 3—figure supplement 1B*). In contrast, the peripheral localization in G1 of late origin *ARS501*, which is not Fkh-activated, was not altered by deletion of *FKH1* (*Figure 3—figure supplement 1C*). These results suggest that *FKH1* plays an expansive role in relocalizing replication origins from the nuclear periphery to the nuclear interior in G1 phase.

## DDK- but not CDK-dependent step of replication initiation drives origin relocalization

Because our previous study indicated that Fkh1/2 was required for origin recruitment of Cdc45 in G1 phase (*Knott et al., 2012*), we tested the requirement for *CDC7*, which encodes the catalytic subunit of *DBF4*-dependent kinase (DDK), and is required for Cdc45 origin-loading

*Figure 2 continued*

replication origins indicated; ARS305$^{V-R}$ resides at the ARS522 (aka: ARS501) locus.

DOI: https://doi.org/10.7554/eLife.45512.004

The following figure supplement is available for figure 2:

**Figure supplement 1.** Normal dosage of Fkh1 is sufficient to advance firing time of ARS305$^{V-R}$.

DOI: https://doi.org/10.7554/eLife.45512.005

(reviewed in *Tanaka and Araki, 2013*). We introduced the *cdc7-as3* allele, the kinase activity of which is inhibited by ATP analog PP1 (*Wan et al., 2006*), and tested whether *FKH1*-induced origin relocalization occurs with inhibition of *CDC7* function. Remarkably, ARS305$^{V-R}$ relocalization was eliminated by inhibition of Cdc7-as3 kinase with PP1 (*Figure 4A*, compare with non-induction in *Figure 1D*). These results suggest that DDK activity is required for origin relocalization in G1-arrested cells.

A role for DDK in G1 phase was unexpected as DDK activity has been reported to be low in α-factor-arrested G1 cells due to instability of Dbf4 (*Nougarède et al., 2000*; *Oshiro et al., 1999*). To provide further evidence for DDK's role, we tested a native origin without *FKH1* overexpression, and to inactivate *CDC7*, we chose the temperature-sensitive *cdc7-4* allele (*Hereford and Hartwell, 1974*). For this experiment, G1-arrested *WT* and *cdc7-4* cells bearing a lacO array inserted near *ARS305* and expressing LacI-GFP were shifted to the non-permissive temperature and *ARS305* location was determined. Compared to *WT* cells, *cdc7-4* cells at the non-permissive temperature showed a significant increase in the proportion of cells with *ARS305* near the nuclear periphery (*Figure 4B*). This result supports the conclusion that DDK activity is required for origin re-positioning in G1 phase cells.

Fkh1-origin binding is cell cycle-regulated, occurring in G1 and S phases (*Ostrow et al., 2014*), suggesting that the requirement for DDK activity in Fkh1-stimulated origin relocalization might be due to dependence of Fkh1 origin-binding on DDK. We tested this possibility by performing chromatin immunoprecipitation analysis of Fkh1 comparing *WT* and *cdc7-as3* cells. The results showed that binding of Fkh1 to *ARS305* and other Fkh-activated origins was largely unaffected by Cdc7 inhibition (*Figure 4—figure supplement 1*). Thus, Fkh1 origin binding appears to be independent of DDK activity, and, by inference, of the subcellular change in localization resulting from DDK inhibition. Alternatively, the requirement for DDK activity in Fkh1-stimulated origin relocalization may reflect Fkh1 acting upstream of DDK, which would comport with a recent report that a critical role of Fkh1 in origin stimulation is DDK recruitment through direct physical interaction with Dbf4 (*Fang et al., 2017*). We tested whether the same mechanism is responsible for Fkh1-induced origin re-positioning by testing the effect on *ARS305* positioning in cells expressing Dbf4 lacking its C-terminus (*dbf4ΔC*), which is required for interaction with Fkh1 (*Fang et al., 2017*). Deletion of *DBF4's* C-terminus had a similar effect on origin localization as *FKH1* deletion, with greater enrichment of *ARS305* near the nuclear periphery (*Figure 4C*, compare with *Figure 3A*), consistent with Fkh1 and Dbf4 acting in the same pathway.

The essential function of DDK in origin firing is phosphorylation of MCM helicase subunits, particularly Mcm4, resulting in removal of auto-inhibition and enabling recruitment of helicase accessory protein Cdc45 through its loading factor Sld3 (reviewed in *Tanaka and Araki, 2013*). To test whether the requirement for Cdc7 kinase activity in origin relocalization reflects its function in Mcm4 helicase phosphorylation, we introduced into the *cdc7-as3* strain an allele of *MCM4*, *MCM4-DD/E(7) +DSP/Q(7)* abbreviated herein as *MCM4-14D*, which contains 14 S/T->D substitutions that mimic critical DDK-phosphorylated residues in Mcm4, and suppresses reduced Cdc7 kinase activity (*Randell et al., 2010*). The presence of *MCM4-14D* restores ARS305$^{V-R}$ relocalization upon *FKH1* induction in the *cdc7-as3* strain inhibited by PP1 (*Figure 5A*). This supports the conclusion that the function of Cdc7 kinase required for origin relocalization is phosphorylation of Mcm4.

We tested whether completion of the DDK-dependent step, that is Sld3 and Cdc45 loading, is required for origin relocalization by testing the effect of inactivation of *CDC45* function. The cold-sensitive *cdc45-1* allele exhibits interdependence with heat-sensitive alleles *cdc7-4* and *dbf4-1* in reciprocal temperature-shift experiments, tightly inhibits replication initiation, and reduces Sld3-origin association in G1 phase (*Aparicio et al., 1999*; *Kamimura et al., 2001*; *Owens et al., 1997*). We synchronized *WT* and *cdc45-1* cells in G1 phase at the permissive temperature and shifted the cultures to semi-permissive temperature while maintaining the G1 arrest. Analysis showed that *ARS305* was more peripherally localized in *cdc45-1* cells at the semi-permissive temperature in G1 phase (*Figure 5B*). As origin binding of Cdc45 and Sld3 is interdependent and Cdc45-1 inactivation

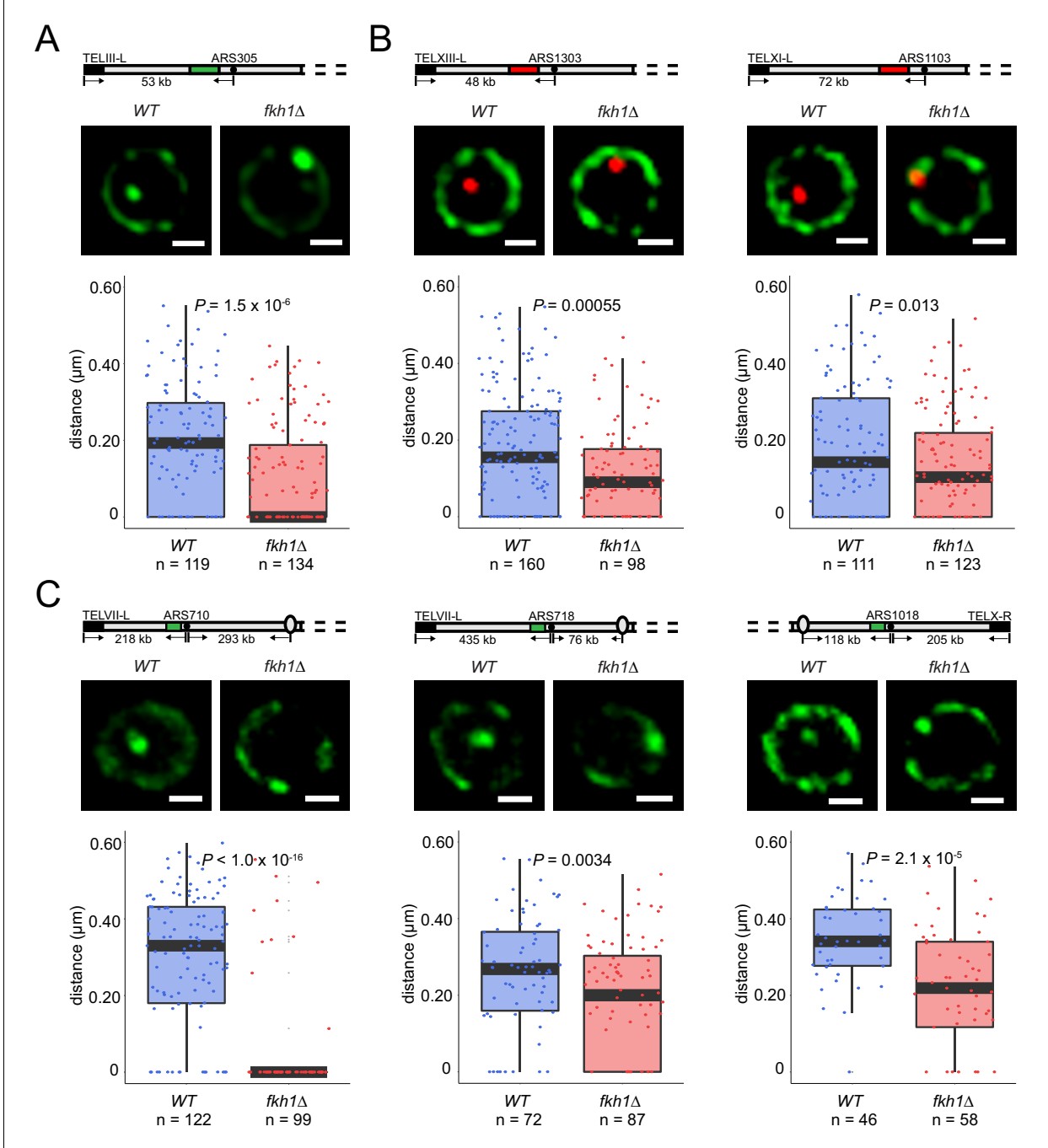

**Figure 3.** Fkh1 determines early origin positioning globally. Diagrams of chromosomes with replication origins labeled with lacO/LacI-GFP (green-filled segment) or tetO/TetR-Tomato (red-filled segment) are shown above the corresponding images. Distances between origins (black-filled spheres) and telomeres, and in some cases centromeres (ovals), are indicated and include ~14 kb or 16 kb added by lacO or tetO repeats, respectively; elements are not drawn to scale. Cells of *WT* and *fkh1Δ* strains with *ARS305-GFP* (HYy151, HYy147) in (**A**), *ARS1303-Tomato* (HYy166, HYy173) and *ARS1103-Tomato* (HYy165, HYy172) in (**B**), *ARS710-GFP* (MPy6, MPy10), *ARS718-GFP* (MPy20, MPy21), and *ARS1018-GFP* (MPy19, MPy22) in (**C**), all expressing *NUP49-GFP,* were arrested in G1 phase at 25°C with 1x α-factor 2 hr and live images were captured; scale bar = 0.5 µm. Distances from origin foci to nuclear periphery were determined, plotted as quartile boxplots, and analyzed by a z-test.

DOI: https://doi.org/10.7554/eLife.45512.006

The following figure supplement is available for figure 3:

**Figure supplement 1.** Fkh1 determines early origin positioning globally.
DOI: https://doi.org/10.7554/eLife.45512.007

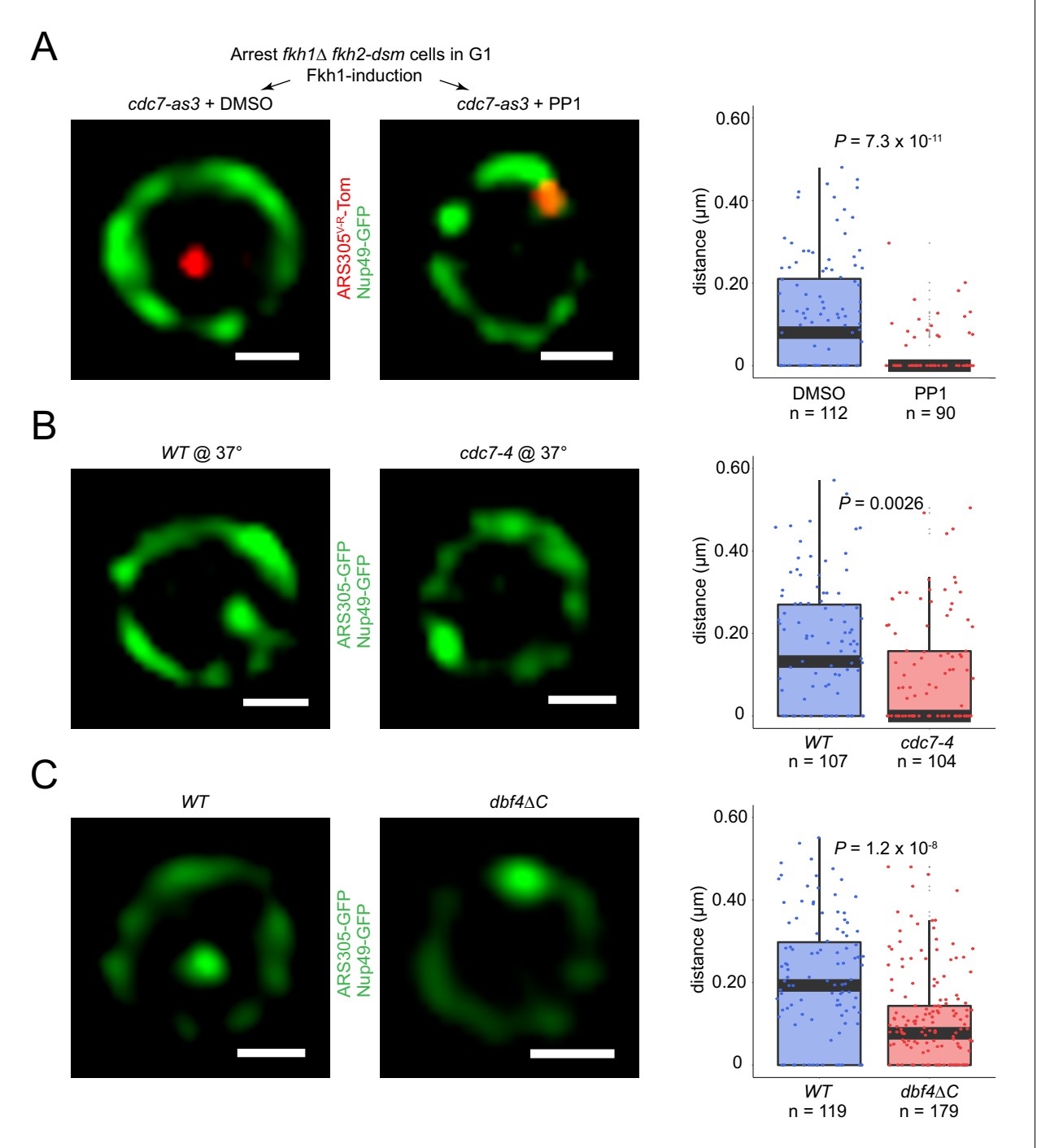

**Figure 4.** Origin localization in G1 is DDK regulated. (**A**) HYy186 (*fkh1Δ fkh2-dsm GAL-FKH1 ARS305^{V-R}-Tomato NUP49-GFP cdc7-as3*) cells were subjected to *FKH1*-induction scheme as described in ***Figure 1C*** legend except that PP1 or DMSO (vehicle) was included with α-factor, and images were captured. (**B**) Cells of *ARS305-GFP NUP49-GFP* strains HYy151 (*WT*) and HYy191 (*cdc7-4*) were arrested in G1 with 1x α-factor 2 hr at 25°C followed by 1 hr incubation at 37°C with 2x α-factor, and images were captured. (**C**) HYy181 (*ARS305-GFP NUP49-GFP dbf4ΔC*) cells were arrested in G1 phase with 1x α-factor 2 hr at 25°C and live images were captured. The control experiment with *WT* cells (HYy151) is shown in ***Figure 3A***. (**A–C**) Scale bar = 0.5 μm. Distances from origin foci to nuclear periphery were determined, plotted as quartile boxplots, and analyzed by a z-test.

DOI: https://doi.org/10.7554/eLife.45512.008

The following figure supplement is available for figure 4:

**Figure supplement 1.** Fkh1 binds origins independently of Cdc7 function.
DOI: https://doi.org/10.7554/eLife.45512.009

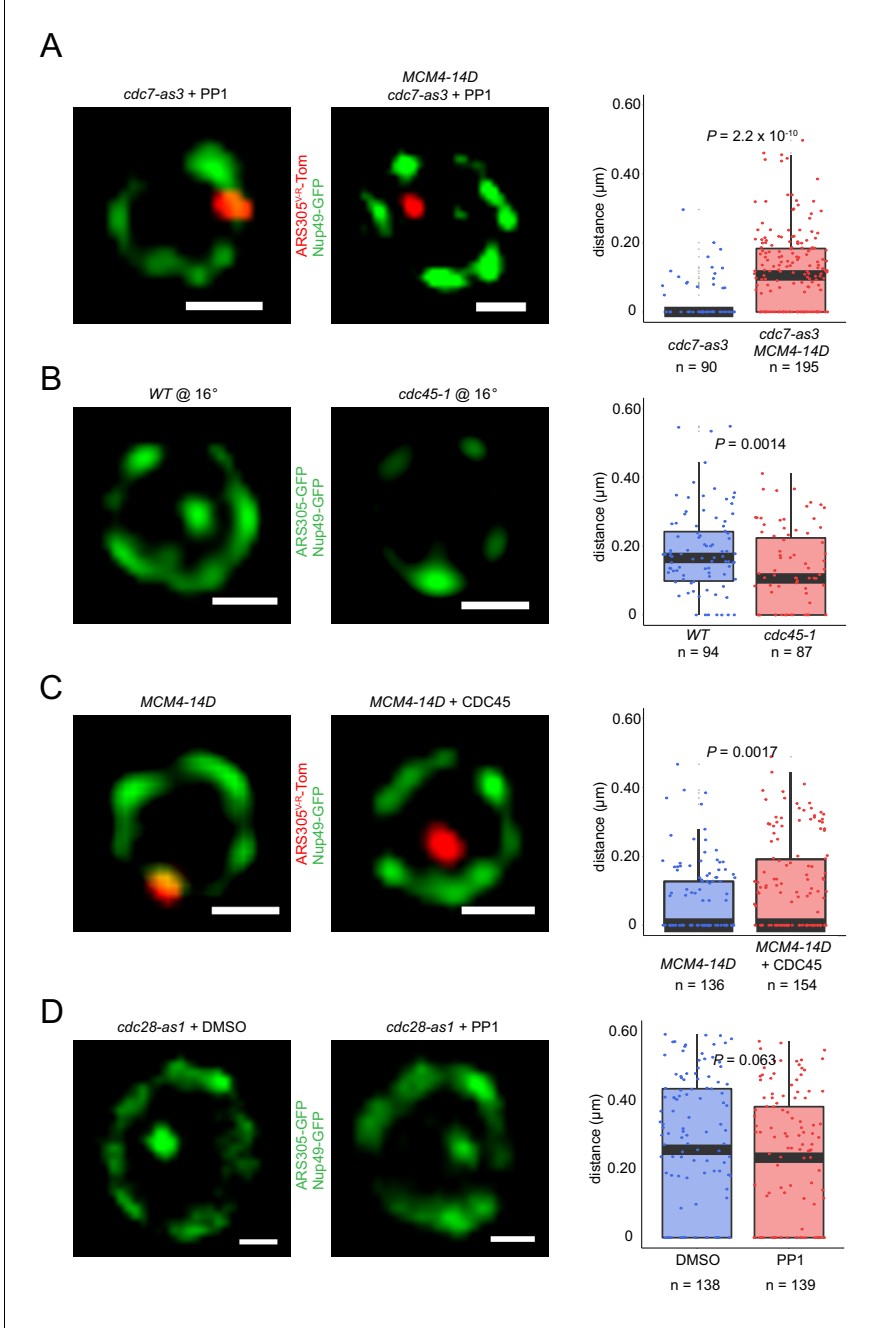

**Figure 5.** DDK regulation of origin localization reflects its phosphorylation of Mcm4 and consequent Cdc45 loading. (**A**) Fkh1-induction scheme with PP1 as described in *Figure 4A* legend was carried out with *fkh1Δ fkh2-dsm GAL-FKH1 ARS305$^{V-R}$-Tomato NUP49-GFP* strains HYy186 (*cdc7-as3*) and HYy177 (*cdc7-as3 MCM4-14D*), and images captured. (**B**) *ARS305-GFP NUP49-GFP* strains HYy151 (*WT*) and HYy184 (*cdc45-1*) were arrested in G1 with 1x α-factor 1 hr at 30°C followed by 2 hr incubation at 16°C with 1x α-factor, and images were captured. (**C**) Cells of strain HYy177 harboring no plasmid or high-copy plasmid expressing *CDC45* were arrested in G1 with 1x α-factor 2 hr at 25°C and images captured. (**D**) *ARS305-GFP NUP49-GFP* strain HYy197 (*cdc28-as1*) cells were arrested in G1 phase with 0.5x α-factor 2 hr at 25°C, PP1 or DMSO was added and incubated one additional hour with 0.5x α-factor, and images were captured. (**A–D**) Scale bar = 0.5 μm. Distances from origin foci to nuclear periphery were determined, plotted as quartile boxplots, and analyzed by a z-test.

DOI: https://doi.org/10.7554/eLife.45512.010

The following figure supplement is available for figure 5:

**Figure supplement 1.** CDK activity is dispensable for origin localization in G1.

DOI: https://doi.org/10.7554/eLife.45512.011

reduces Sld3-origin binding (*Kamimura et al., 2001*), these results suggest that assembly of Sld3-Cdc45 onto origins is required for origin relocation.

Cdc45 is incorporated into replisomes as a component of the active helicase complex together with MCM and GINS. However, Cdc45 is present in low abundance and is likely limiting for the total number of active replisomes that may be simultaneously active (*Mantiero et al., 2011*; *Tanaka et al., 2011*). We noticed that the presence of Mcm4-14D was not sufficient to relocalize *ARS305^{V-R}* in *fkh1Δ fkh2-dsm* cells in the absence of Fkh1 induction (*Figure 5C*), which might be contrary to expectations if the only function of Fkh1 is to physically recruit DDK, which has been rendered dispensable by Mcm4-14D. We note, however, that *MCM4-14D* does not suppress a deletion of *CDC7* (S.P. Bell, personal communication) suggesting that residual DDK activity is required for sufficient origin firing, and hence, Fkh1 may act to target this residual activity to specific origins. Absent this targeting, we postulated that the limited abundance of Cdc45 would be further diluted amongst all licensed origins due to potentiation by Mcm4-14D. To test this idea, we introduced a high-copy plasmid expressing Cdc45 from its native promoter into the *fkh1Δ fkh2-dsm MCM4-14D* strain, and examined origin location. Consistent with the notion that Cdc45 is limiting for execution of the DDK-dependent step, expression of high copy Cdc45 significantly increased the frequency of *ARS305* more distal from the periphery (*Figure 5C*). This finding supports the conclusion that full execution of the DDK-dependent step in the form of Cdc45 loading, as opposed to Mcm4 phosphorylation itself or phosphorylation of other targets is required for origin relocation.

As the interior localization of early origins occurs in α-factor-arrested, G1 phase cells, cyclin-dependent kinase (CDK) activity would appear to be dispensable because G1 phase cells have very low levels of S/G2/M-CDK activities, and G1-CDKs, which are required for passage through Start, are inhibited by α-factor (reviewed in *Mendenhall and Hodge, 1998*). Nevertheless, low levels of S/G2/M-CDK activities in G1 phase cannot be ruled out, and indeed, it appears that low levels of DDK are involved. Thus, to address the possibility that CDK activity might be contributing to G1 phase origin dynamics, we tested the requirement for *CDC28*, the Cdk1 kinase, using analog-sensitive *cdc28-as1* cells (*Bishop et al., 2000*). In G1-arrested cells, inhibition of Cdc28-as1 activity with PP1 did not alter localization of *ARS305* (*Figure 5D*), although budding was inhibited upon release from α-factor arrest indicating effective inhibition of Cdc28-as1 (*Figure 5—figure supplement 1A*). Similarly, PP1 treatment of cycling *cdc28-as1* cells did not alter distribution of *ARS305* in G1 phase cells (*Figure 5—figure supplement 1B*), while DNA content analysis showed delayed entry of cells into S phase indicating effective inhibition of Cdc28-as1 (*Figure 5—figure supplement 1C*). Thus, CDK activity appears to be dispensable for normal, Fkh1-dependent positioning of *ARS305*. Overall, our findings indicate that DDK but not CDK activity stimulates origin relocation in G1 phase.

## Origin mobility increases with origin relocation

Fkh1 might facilitate origin relocation by promoting origin mobilization (release from the periphery or movement per se), or by increasing the stability of origin-origin interaction after relocation. In addition to changes in location, replication origins exhibit decreased rate of mobility during progression into S phase (*Heun et al., 2001b*). We directly investigated how Fkh1 affects origin mobility by tracking the locations of *ARS305* and *ARS718* in individual *WT* and *fkh1Δ* cells over time, and applying mean square displacement (MSD) analyses (*Marshall et al., 1997*). The analysis shows significantly lower plateau of MSD curves in *fkh1Δ* cells (*Figure 6A*), consistent with less nuclear space explored. Calculation of the radius of constraint (Rc) and the corresponding volume of space explored reveals that *ARS305* explores about 2.5-fold more volume and *ARS718* explores about 3.8-fold more volume in *WT* than *fkh1Δ* cells. Tracings of the paths of origin foci show confinement proximal to the nuclear periphery in *fkh1Δ* cells (*Figure 6B*). Together, these data show that Fkh1 stimulates origin mobilization.

## Discussion

This study reveals new links between key molecular interactions in replication initiation and the localization and mobility of replication origins within the nucleus. In particular, we show that early origin specification in G1 phase by Fkh1 induces a change from peripheral to interior nuclear localization of Fkh1-activated origins. Quite remarkably, we find that origin relocation requires execution of the DDK-dependent step of origin firing that loads Cdc45. That the DDK requirement reflects the key,

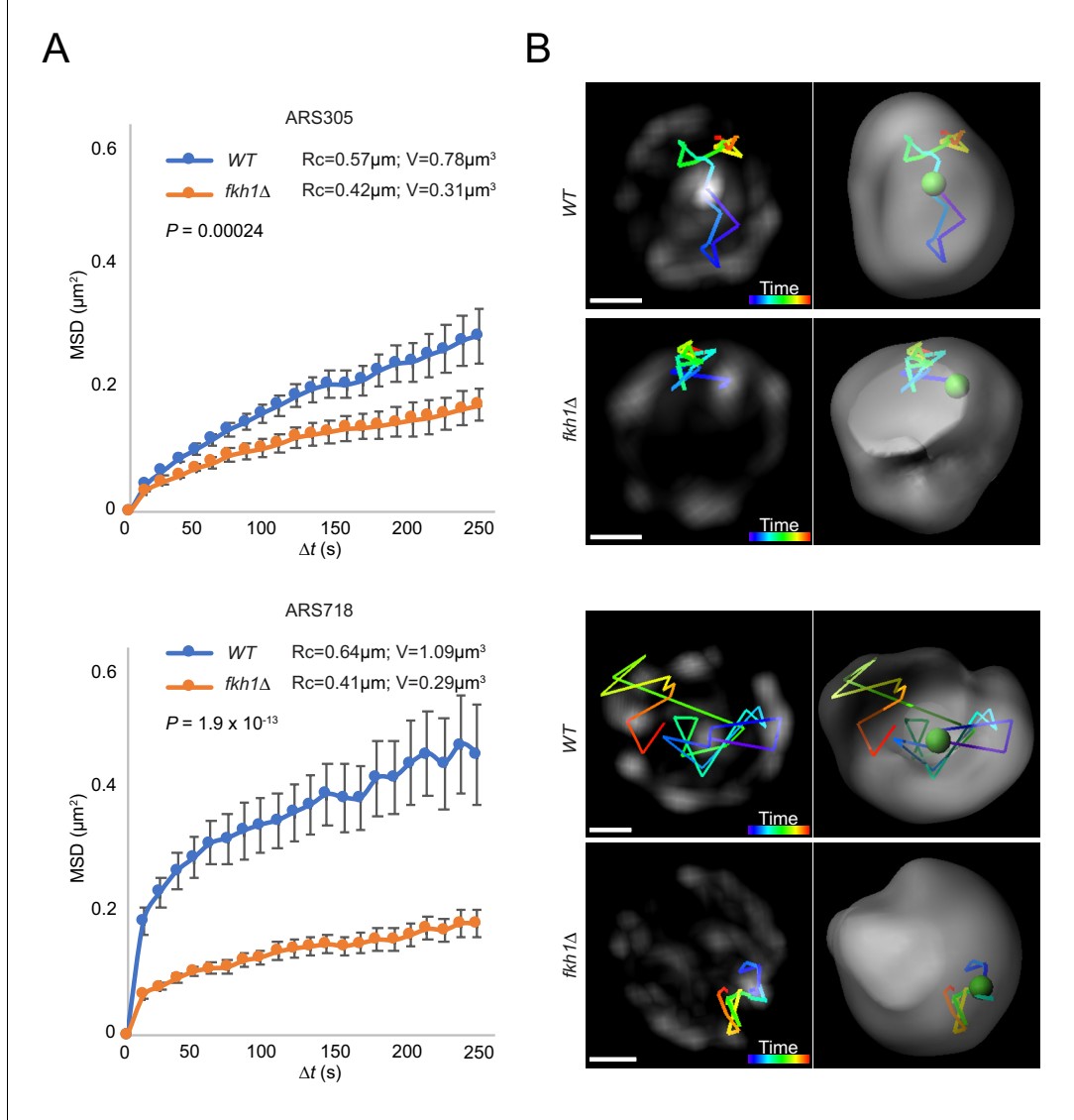

**Figure 6.** Origin mobility increases with origin relocalization. (**A**) Mean-squared-displacement (MSD) analysis of tracking data for *ARS305-GFP* strains HYy151 (*WT*) and HYy147 (*fkh1Δ*) *and ARS718-GFP* strains MPy20 (*WT*) and MPy21 (*fkh1Δ*). Radius of constraint (Rc) and volume searched (**V**) are given, and statistical significance comparing *WT* and *fkh1Δ* was estimated by two-tailed Mann-Whitney test. (**B**) Images (left) and 3D reconstructions with Imaris (right) showing examples of tracks of origin focus over time (color corresponding to time progression); scale bar = 0.4 μm. Movies of the individual *ARS305* time-courses are available as *Figure 6—video 1* and *Figure 6—video 2*.

DOI: https://doi.org/10.7554/eLife.45512.012

The following videos are available for figure 6:

**Figure 6—video 1.** Origin tracking in WT.

DOI: https://doi.org/10.7554/eLife.45512.013

**Figure 6—video 2.** Origin tracking in fkh1Δ.

DOI: https://doi.org/10.7554/eLife.45512.014

recognized function of DDK in replication initiation, that is, phosphorylation of MCM proteins leading to Sld3-Cdc45 origin-loading, is demonstrated by the bypass of *CDC7* requirement by phospho-mimetic mutations in *MCM4-14D* as well as the dependence on *CDC45* function. This early execution of the DDK step was unexpected because DDK levels have been reported to be very low in α-factor-arrested G1 cells due to Dbf4 instability (*Nougarède et al., 2000*; *Oshiro et al., 1999*). Our findings provide direct evidence that DDK is active in G1 phase and has already established origin timing by late G1 phase in α-factor arrest. This finding explains previous observations that Sld3 and

Cdc45 associate with early replication origins in G1 phase (*Aparicio et al., 1999*; *Kamimura et al., 2001*). Our findings are also consistent with a more recent study showing that Sld3- and Cdc45-origin association in G1 phase is DDK-dependent and CDK-independent, as well as the conclusion that DDK acts prior to and independently of S-CDK (*Heller et al., 2011*; *Yeeles et al., 2015*), the latter of which is dispensable for the observed origin relocalization.

The finding that Fkh1 and DDK are required for origin relocalization fits well with the recent finding that Fkh1 acts to stimulate origin firing by directly recruiting Dbf4 through physical interaction (*Fang et al., 2017*), and extends our understanding of the significance of this interaction to replication initiation *via* nuclear positioning of replication sites. As predicted by this interaction model, inactivation of DDK activity should phenocopy deletion of *FKH1*, as we have herein demonstrated with depletion of *CDC7* function. Moreover, specific deletion of Dbf4's C-terminus, which is required for interaction with Fkh1, also phenocopies deletion of *FKH1*. Furthermore, the absence of *FKH1* and *FKH2* function is bypassed by the *MCM4-14D* allele in the presence of increased levels of Cdc45. Together, these results support a mechanism involving Fkh1 recruitment of DDK activity to load Cdc45 at a subset of origins in G1, corresponding with a change in subnuclear positioning of these origins, and early firing in the subsequent S phase.

Chromosome conformation capture (Hi-C) experiments have indicated that early firing replication origins preferentially interact with each other, or 'cluster' in G1 phase (*Duan et al., 2010*). Related studies have shown that Fkh1/2 is required for these spatial interactions amongst early origins (*Eser et al., 2017*; *Knott et al., 2012*; *Ostrow et al., 2017*). We propose that the origin clustering interactions revealed by Hi-C experiments directly reflect origin localization to distinct nuclear territories as observed microscopically. Thus, localization to the nuclear interior might increase the likelihood for physical interaction amongst this subset of origins. Such interactions may be driven by cooperative interactions between Fkh1-bound origins recruiting limiting initiation factors such as Dbf4, Sld3 and Cdc45. This aggregation of origins selected for early/efficient activation has the inevitable consequence that replication initiation transforms these origin clusters into replication foci, which have been observed as concentrations of DNA synthesis and replication factors (*Berezney et al., 2000*; *Frouin et al., 2003*; *Hozák et al., 1994*; *Kitamura et al., 2006*; *Nakamura et al., 1986*; *Newport and Yan, 1996*). These assemblages may contribute to efficient chromosomal replication initiation and elongation in multiple ways, such as accretion of activities and co-factors directly required for DNA synthesis (e.g.: dNTP production), and scaffolding to co-localize and coordinate replication with related activities like chromatin assembly, cohesion establishment, topological resolution, and DNA repair.

It remains to be determined exactly what maintains either the peripheral or interior localization of origins or what drives relocalization between different subnuclear zones. While telomere tethering to the nuclear envelope has been assumed to cause the peripheral localization of telomere-proximal origins, we find that early origins distal from telomeres that are normally enriched in the nuclear interior, are closer to the nuclear periphery in cells lacking Fkh1, suggesting that perinuclear localization represents a default state for most origins irrespective of telomere tethering (*Figure 7A*). It is unclear what promotes this origin localization. Complete elimination of origin function results in even more peripheral distribution of the locus suggesting that peripheral localization is independent of an origin tethering mechanism, and that interior localization is linked to activation of origin function, which may occur, with less efficiency, in the absence of Fkh1/2. There may be passive exclusion from the interior where other activities like transcription may predominate in early G1, or there may be a dedicated tethering mechanism, though origin association with the periphery does not appear to be as stringently localized or as stable as that of telomeres (*Hediger et al., 2002*; *Heun et al., 2001a*; *Heun et al., 2001b*).

In addition to subtelomeric origins, Rif1 regulates and associates with origins distal from telomeres and with the nuclear envelope, and therefore could potentially tether origins to the periphery (*Figure 7A*) (*Hafner et al., 2018*; *Park et al., 2011*; *Peace et al., 2014*). Rif1 interacts with Dbf4 and with the counteracting PP1 phosphatase, suggesting that the Rif1-origin interaction may be downregulated by DDK-dependent phosphorylation of MCM proteins and/or Rif1 (*Davé et al., 2014*; *Hiraga et al., 2014*; *Mattarocci et al., 2014*). Thus, Fkh1-mediated, origin-specific recruitment of DDK may overwhelm Rif1-mediated PP1 inhibition locally and thereby release the origin from peripheral tethering (*Figure 7B*). Consequent Cdc45 loading might effectively prevent reversal of MCM phosphorylation and fully disrupt interaction with Rif1-PP1. Alternatively, MCM

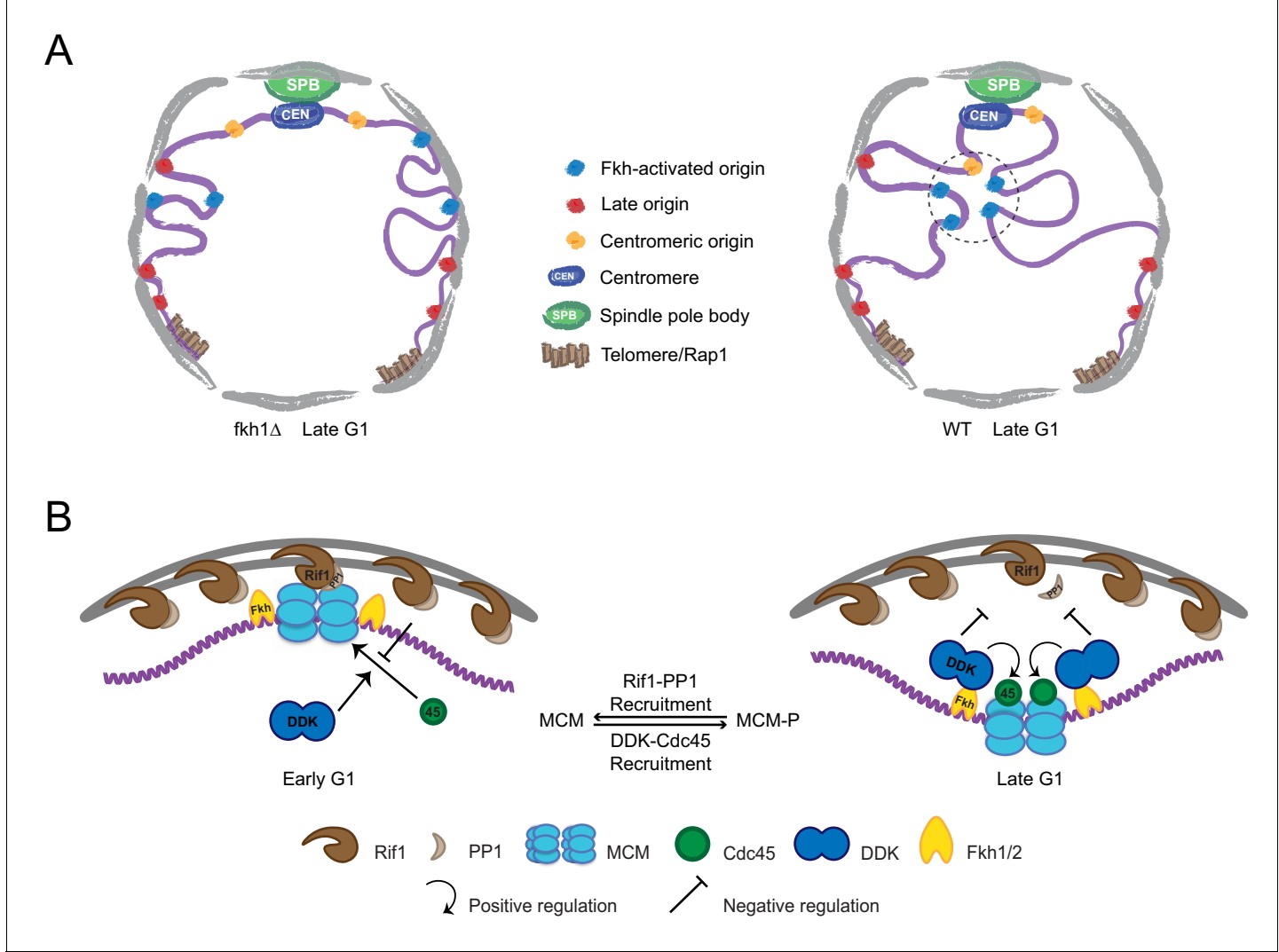

**Figure 7.** Model of origin localization linked to initiation. (**A**) Absent Fkh1, most replication origins are enriched at nuclear periphery, however, Fkh1 binding to a subset of origins allows execution of the DDK-dependent step of initiation, resulting in release from the nuclear periphery and/or capture in the nuclear interior to form early origin clusters. (**B**) Hypothetical mechanism for origin tethering to the nuclear periphery regulated by Rif1-PP1 versus Fkh1-DDK-Cdc45 activities. Rif1 associates with inner nuclear membrane and with licensed replication origins, while associated PP1 antagonizes execution of the DDK-dependent step. Fkh1-dependent recruitment of DDK results in phosphorylation of MCM, Cdc45 loading and local release from Rif1 and PP1. See text for further discussion.

DOI: https://doi.org/10.7554/eLife.45512.015

phosphorylation and/or Cdc45 loading might change the licensed origin's biophysical properties, thereby forcing the origin to occupy and search different space and/or capture scaffolding factors, which may themselves be localized to the interior and thus stabilize interior localization. Future studies aimed at more detailed examination of how individual factors affect origin mobility should provide further insights.

Previous studies have concluded that peripheral localization of origins is neither necessary nor sufficient to regulate initiation timing. In one study, subtelomeric origin *ARS501* remained late firing following excision (in α-factor-arrested G1 cells) from the chromosome, which allowed its diffusion away from the nuclear periphery, leading the authors to suggest that peripheral localization might promote a chromatin mark that maintains late timing (*Heun et al., 2001a*). However, we have shown that induction of Fkh1 (in α-factor-arrested cells) can reprogram timing of a Fkh-activated origin inserted into the *ARS501* locus (*Peace et al., 2016*). We propose that the relevant chromatin mark is

MCM phosphorylation, removal of which is promoted by peripheral localization and addition by DDK recruitment. Thus, excised *ARS501* remains late despite its mobilization because DDK is limiting and already recruited by Ctf19 and Fkh1/2 to other origins. Other studies have shown that tethering of early origins to the nuclear periphery does not delay their activation (*Ebrahimi et al., 2010*; *Zappulla et al., 2002*). However, both origins in these previous studies, *ARS305* and *ARS607*, are Fkh-activated origins that we have shown can overcome the replication initiation delay associated with peripheral localization. Overall, these previous findings fit neatly into our model, which suggests that interior origin localization is a consequence rather than a cause of early timing.

## Materials and methods

### Plasmid constructions

Plasmids are listed in *Supplementary file 1*. Plasmids were constructed using Gibson Assembly kit (SGI cat#GA1200) unless otherwise indicated. Restriction enzymes, T4 DNA ligase, and Klenow were from New England Biolabs and used according to their protocols. Mutagenesis was carried out using QuikChange Lightning Multi kit (Agilent cat#210515); sequence changes were confirmed by DNA sequencing (Retrogen Inc). STBLII cells were used for maintenance of plasmids containing tandem repeats (Invitrogen cat#10268019). Primer sequences for plasmid constructions are given in *Supplementary file 2*. NUP49-GFP was PCR-amplified from pUN100-GFP*-Nup49 (from V Doye) using primers Nup49-GFP-F and Nup49-GFP-R and subcloned into *Xhol+Sacl* digested vectors pRS403 and pRS404 (*Sikorski and Hieter, 1989*) to yield p403-Nup49-GFP and p404-Nup49-GFP, respectively. Primers ADE2-up-F and ADE2-int-R and separately ADE2-farup-F and ADE2-up-R were used to amplify sequences of *ADE2* for targeting and as a selectable marked; these were inserted into pbluescriptKS+ to create pblueKS-ADE2target. TetR-Tomato was PCR-amplified from plasmid p402-TetR-Tomato (from S Sabatinos) using primers TetR-Tom-F and TetR-Tom-R and inserted into *PacI*-digested pblueKS-ADE2target to generate pTetR-Tom-ADE2. 2.1kbp *KpnI-SacI* fragment containing LacI-GFP was subcloned from pAFS135 (from J Bachant) into pRS404 digested with same enzymes to create p404-LacI-GFP. Plasmids containing tetO (pGS004 from J Bachant) or lacO (pJBN164 from J Bachant) arrays were modified by introduction of genomic sequences to target integration near different origins. The following primer pairs were used to generate sequences adjacent to the indicated origins (with the corresponding chromosomal coordinates given in parentheses): primers ARS501-tetO-F and ARS501-tetO-R for *ARS501* (V:547812–548329), primers ARS1103-tetO-F and ARS1103-tetO-R for *ARS1103* (XI:54673–54996) and primers ARS1303-tetO-F and ARS1303-tetO-R for *ARS1303* (XIII:31983–32247), and these were inserted into *KpnI+ClaI* digested pGS004 yielding pARS501-tetO, pARS1103-tetO, and pARS1303-tetO, respectively. Likewise, the following primer pairs were used to generate sequences adjacent to the indicated origins: primers ARS710-lacO-F and ARS710-lacO-R for *ARS710* (VII:204305–204831), primers ARS718-lacO-F and ARS718-lacO-R for *ARS718* (VII:422375–423281), and primers ARS1018-lacO-F and ARS1018-lacO-R for *ARS1018* (X:539662–540395), and these were inserted into *XhoI+KpnI* digested pJBN164 yielding pARS710-lacO, pARS718-lacO, and pARS1018-lacO, respectively. Two adjacent regions near *ARS305* were PCR-amplified using primer pair NotI-ARS305-5' and XhoI-ARS305-5' (III:37283–37778) and primer pair NotI-ARS305-3' and KpnI-ARS305-3' (III:37779–38282) and digested with *NotI* and *XhoI* and *NotI* and *KpnI*, respectively; these fragments were ligated into pRS404 digested with *XhoI* and *KpnI*. The *XhoI-KpnI* fragment was subcloned by digestion and ligation into pJBN164 digested with same enzymes to yield pARS305-lacO. Plasmids p501Δ-ARS305-ΔACS and p501Δ-ARS305-Δ2BS were created by mutagenesis of p501Δ-ARS305 (*Peace et al., 2016*) with primers ARS305-ΔACS-mut1, ARS305-ΔACS-mut2 and ARS305-Δ2BS-mut1, and ARS305-Δ2BS-mut2, respectively. Plasmid p404-ars305Δ-BInc was constructed as described for p306-ars305Δ-BrdU-Inc (*Zhong et al., 2013*) except that p404-BrdU-Inc (*Viggiani and Aparicio, 2006*) was used instead of p306-BrdU-Inc. p404-ars305Δ-BInc was digested with *PmlI* and *KpnI*, blunted-ended with Klenow, and ligated with T4 DNA ligase to remove the *TRP1* selectable marker, yielding p400-ars305Δ-BInc. The 1.5 kb *SalI-SpeI* fragment containing the KanMx cassette from pFA6-KanMx (*Longtine et al., 1998*) was ligated into *SalI-SpeI*-digested p400-ars305Δ-BInc, creating pKanMx-ars305Δ-BInc. The *cdc45-1* allele was PCR-amplified from strain YB298 (from B Stillman) with primers Cdc45-F and Cdc45-R and inserted into *SacI+KpnI* digested pRS406 (*Sikorski and Hieter, 1989*) to create p406-cdc45-1.

## Yeast strain constructions

All strains are congenic with SSy161, derived from W303-1a (*RAD5*) (*Viggiani and Aparicio, 2006*); complete genotypes are given in *Supplementary file 3*. Strain constructions were carried out by genetic crosses or lithium acetate transformations with linearized plasmids or PCR products generated with hybrid oligonucleotide primers having homology to target loci (*Ito et al., 2001*; *Longtine et al., 1998*); primer sequences for strain constructions are given in *Supplementary file 2*. Genomic alterations were confirmed by PCR analysis or DNA sequence analysis as appropriate.

*FKH1* was deleted using primers Fkh1-up and Fkh1-down to amplify KanMx selectable marker from pFA6-KanMx (*Longtine et al., 1998*). *FKH2* was replaced by *fkh2-dsm* in two steps: first, *FKH2* was entirely replaced with *URA3* (*C. albicans*) using pAG61 (Addgene), and the resulting strain was transformed with *fkh2-dsm* DNA from p405-fkh2-dsm (*Ostrow et al., 2017*) followed by selection on 5-FOA. GAL-FKH1 was introduced using p405-GAL-FKH1 and *FKH1* was FLAG-tagged as described previously (*Peace et al., 2016*). *ARS501* was replaced by *ARS305* or mutant versions of *ARS305* by transformation with p501Δ-ARS305, p501Δ-ARS305-ΔACS, or p501Δ-ARS305-Δ2BS as described previously (*Peace et al., 2016*). BrdU incorporation cassette was introduced, replacing *ARS305*, by transformation with BglII-digested p404-ARS305-BrdUInc. The *cdc28-as1* allele was introduced by pop-in/pop-out of plasmid pJUcdc28-as1 digested with HindIII. The *cdc7-as3* allele was introduced as described previously (*Zhong et al., 2013*); *cdc7-4* was back-crossed from H7C4A1 (from L Hartwell) into the W303 background four times, with the final cross to HYy151. *MCM4-DD/E+DSP/Q* (referred to in text as *MCM4-14D*) was introduced by transformation with PacI-digested pJR179 (from SP Bell). The *cdc45-1* allele was introduced by crossing with strain YB298 (from B Stillman) or by pop-in/pop-out with BglII-digested p406-cdc45-1. The *dbf4ΔC* allele was constructed by insertion of a non-sense codon with the KanMx cassette from pFA6-KanMx (*Longtine et al., 1998*) using primers Dbf4-up and Dbf4-down. TetR-Tomato was introduced by transformation with PacI-digested pTetR-Tom-ADE2. LacI-GFP was introduced by transformation with HindIII-digested p404-LacI-GFP. The tetO or lacO arrays were introduced by transformation with pARS501-tetO, pARS1103-tetO, pARS1303-tetO, pARS305-lacO, pARS710-lacO, pARS718-lacO and pARS1018-lacO digested with PacI, PshAI, BlpI, NotI, PshAI, SnaBI, and BlpI, respectively.

## Cell growth and synchronization

Cells were grown at 25°C unless otherwise indicated. For microscopy, cells were grown in complete synthetic medium supplemented with 15 μg/mL adenine (CSM+ade) +2% dextrose, unless otherwise indicated (raffinose or galactose); for QBU and ChIP-seq, cells were grown in YEP +2% dextrose, unless otherwise indicated (raffinose or galactose). G1 arrest was achieved by incubation with 2.5 nM (1x) α–factor (Sigma T6901); for most extended arrests, a fresh or additional dose of α–factor was added at time of induction/non-induction or at time of temperature shift as indicated in figure legend. PP1 (Tocris Biosciences) was added to 25 μM at the time of initial α–factor incubation. Reagents are listed in *Supplementary file 4*.

## Live-cell fluorescence microscopy and image analysis

~$5 \times 10^6$ cells were harvested by centrifugation and spread on agarose pads made of CSM+ade +4% dextrose. A DeltaVision wide-field deconvolution microscope was used to capture 28 Z-stacks in 0.25 μm increments for each image. SoftWorX software (Applied Precision/GE Healthcare) was used for deconvolution and three-dimensional reconstruction of nuclei, and for measuring the distance between replication origins and nuclear periphery. For experiments with mutant strains having irregularly shaped nuclei (e.g.: *fkh1Δ fkh2-dsm*), measurements were made in three-dimensions; otherwise, measurements were made in two dimensions using a few middle sections as previously described (*Ryu et al., 2015*). A z-test was applied to compare the distribution of measured distances. Images are max intensity projections of two to four middle Z-stacks.

## Quantitative BrdU Immunoprecipitation (QBU)

QBU and analysis of sequencing reads was performed as described previously using KAPA Hyper Prep Kit (KK8504) (*Haye-Bertolozzi and Aparicio, 2018*). Data analysis was performed using 352 replication origins classified as Fkh-activated, Fkh-repressed, or Fkh-unregulated (*Knott et al.,*

*2012*); the latter two classes are grouped together as 'other origins' in *Figure 2—figure supplement 1*. Reagents are listed in *Supplementary file 4*.

## Chromatin immunoprecipitation analyzed by sequencing (ChIP-seq)

ChIP-seq and analysis of sequencing reads was performed as described previously using KAPA Hyper Prep Kit (KK8504) (*Ostrow et al., 2015*). Data analysis was performed using 95 replication origins classified as Fkh-activated (*Knott et al., 2012*). Reagents are listed in *Supplementary file 4*.

## Time-lapse video and MSD analysis

A DeltaVision wide-field deconvolution microscope was used to capture 20 Z-stacks in 0.30 μm increments for each time point. GFP signals were imaged every 12 s for 5 min, with 0.1 s exposure for each Z-stack and 32% of transmitted light using an LED source. All time-lapse movies were deconvolved using SoftWoRx. At least 20 individual cells with nearly stationary nuclei were used to track the trajectory of origin focus for each strain using Imaris (Bitplane), and MSD curves, Rc, and volumes were derived as previously described (*Caridi et al., 2018*); the error bars represent standard error.

## Supplemental data files

Raw data, images, and analysis files are available and organized into folders corresponding to main and supplementary figures. Data is available at Dryad (https://dx.doi.org/10.5061/dryad.7bm444s).

## Acknowledgements

We thank J Bachant, SP Bell, V Doye, J Li, S Sabatinos, and B Stillman for providing strains and/or plasmids. We are grateful to C Caridi and SL Forsburg for microscopy support and helpful advice, S Gasser for helpful discussion, and C Caridi and M Michael for critical reading of the manuscript. For assistance with high-throughput DNA sequencing, we thank Daniel Campo, Director of USC's UPC Genome and Cytometry Core. This work was supported by NIH R01-GM05494 (to OMA) and NIH R01-GM117376 (to IEC).

## Additional information

### Funding

| Funder | Grant reference number | Author |
|---|---|---|
| National Institute of General Medical Sciences | R01-GMS 05494 | Oscar M Aparicio |
| National Institute of General Medical Sciences | R01-GMS 117376 | Irene E Chiolo |

The funders had no role in study design, data collection and interpretation, or the decision to submit the work for publication.

### Author contributions

Haiyang Zhang, Resources, Data curation, Formal analysis, Validation, Investigation, Visualization, Writing—original draft, Writing—review and editing; Meghan V Petrie, Resources, Data curation, Formal analysis, Validation, Investigation, Visualization, Writing—review and editing; Yiwei He, Resources, Data curation, Formal analysis, Validation, Investigation, Visualization; Jared M Peace, Resources, Investigation; Irene E Chiolo, Conceptualization, Formal analysis, Supervision, Writing—review and editing; Oscar M Aparicio, Conceptualization, Supervision, Funding acquisition, Investigation, Writing—original draft, Project administration, Writing—review and editing

### Author ORCIDs

Irene E Chiolo (iD) http://orcid.org/0000-0002-3080-550X
Oscar M Aparicio (iD) https://orcid.org/0000-0002-5591-0277

**Decision letter and Author response**
Decision letter https://doi.org/10.7554/eLife.45512.024
Author response https://doi.org/10.7554/eLife.45512.025

## Additional files

### Supplementary files
• Supplementary file 1. Plasmid list.
DOI: https://doi.org/10.7554/eLife.45512.016

• Supplementary file 2. Primer list.
DOI: https://doi.org/10.7554/eLife.45512.017

• Supplementary file 3. Strain list.
DOI: https://doi.org/10.7554/eLife.45512.018

• Supplementary file 4. Reagent list.
DOI: https://doi.org/10.7554/eLife.45512.019

• Transparent reporting form
DOI: https://doi.org/10.7554/eLife.45512.020

### Data availability

Imaging quantification, statistical analysis, sequencing data and MATLAB scripts have been deposited at Dryad (https://dx.doi.org/10.5061/dryad.7bm444s).

The following dataset was generated:

| Author(s) | Year | Dataset title | Dataset URL | Database and Identifier |
|---|---|---|---|---|
| Zhang H, Petrie M, He Y, Peace J, Chiolo I, Aparicio O | 2019 | Data from: Dynamic relocalization of replication origins by Fkh1 requires execution of DDK function and Cdc45 loading at origins in S. cerevisiae | https://dx.doi.org/10.5061/dryad.7bm444s | Dryad Digital Repository, 10.5061/dryad.7bm444s |

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
