## [Decision Letter]

Thank you for submitting your article "Dynamic relocalization of replication origins by Fkh1 requires execution of DDK function and Cdc45 loading at origins" for consideration by *eLife*. Your article has been reviewed by three peer reviewers, including Bruce Stillman as the Reviewing Editor and Reviewer #1, and the evaluation has been overseen by Jessica Tyler as the Senior Editor. The following individual involved in review of your submission has agreed to reveal their identity: Philippe Pasero (Reviewer #2).

The reviewers have discussed the reviews with one another and the Reviewing Editor has drafted this decision to help you prepare a revised submission.

The paper by Zhang et al. addresses a mechanism for the activation of a select subset of origins of DNA replication in the yeast *S. cerevisiae* that are dependent on the Fkh1/2 proteins for the normal timing of their activation during S phase. The authors show that in the absence of Fkh1/2, an origin of DNA replication associates with the nuclear periphery and upon expression of Fkh1, the origin re-localizes to the nuclear interior. The later movement is dependent upon DDK (Cdc7-Dbf4) kinase and Cdc45 activities, but not CDK. They present a plausible model in which Fkh1/2 recruits the rate limiting Dbf4 DDK subunit to the nuclear periphery by biding to Fkh1/2, countering the Rif1-PP1 dephosphorylation of Mcm4 and releasing the origin to the interior of the nucleus where Cdc45 binding prevents its re-localization to the nuclear periphery.

The paper generally is well presented and for the most part the data supports the conclusions, however, the reviewers have raised some points that could be addressed. These are listed below. Thus we ask that a revised paper be submitted for publication in *eLife*.

Essential revisions:

1) The correlation between release from the nuclear periphery and early replication timing leads to the reasonable speculation that sub-nuclear localization mechanistically regulates replication timing. Previous work has concluded that peripheral localization is neither necessary (Heun et al., 2001 https://www.ncbi.nlm.nih.gov/pubmed/11266454) nor sufficient (Zappulla et al, 2002 https://www.ncbi.nlm.nih.gov/pubmed/12062049) to regulate replication timing. Furthermore, earlier work from the Gasser and Brewer & Fangman labs suggested that an epigenetic mark is established at late origins in G1 through their interaction with the nuclear periphery and is maintained later on to determine their timing of initiation, regardless of their subnuclear organization. This model is not mutually exclusive with the one proposed by the authors and should be discussed. This should be addressed in the paper.

2) The view that the re-localization of Fkh1 origins does not depend on residual CDK activity in G1 is not fully supported by the experiment performed with clb5 clb6 mutants. Indeed, S-phase onset is only delayed by 30 minutes in this mutant. In principle, Clb3 and Clb4 could also support a residual CDK activity in alpha-factor arrested cells. It would have been more convincing to use a conditional *cdc28* mutant.

3) A question that the authors do not address is why, in Figure 4—figure supplement 1 does Fkh1 bind non-Fkh1-activated origins? Or, put another way, why does Fkh1 not activate all the origins to which it binds? The manuscript focuses on Fkh1-regulated origins, but it would be important to show that the localization of late origins and Fkh1-independent origins does not depend on Fkh1 and/or DDK activity in G1.

4) The authors' model is difficult to integrate into a more comprehensive model of sub-nuclear localization. In *fkh1∆ fkh2∆,* are all origins at the nuclear periphery? If so, is there any DNA in the nuclear interior? Why is *ARS305^V-R^*-∆ACS confined to the nuclear periphery? In authors' model is that origins bind to the periphery unless released by DDK, but in the case of *ARS305^V-R^*-∆ACS, it seems that the origin is required to drive the locus away from the periphery. Why would a non-origin locus bind to periphery? Do all non-origin loci bind to the periphery? These questions should be addressed by testing if the ARS305 ACS is required for FKH1-dependent localization at its endogenous locus.

---

## [Author Response]

Essential revisions:

*1) The correlation between release from the nuclear periphery and early replication timing leads to the reasonable speculation that sub-nuclear localization mechanistically regulates replication timing. Previous work has concluded that peripheral localization is neither necessary (*Heun et al., 2001 *https://www.ncbi.nlm.nih.gov/pubmed/11266454) nor sufficient (Zappulla et al, 2002 https://www.ncbi.nlm.nih.gov/pubmed/12062049) to regulate replication timing. Furthermore, earlier work from the Gasser and Brewer & Fangman labs suggested that an epigenetic mark is established at late origins in G1 through their interaction with the nuclear periphery and is maintained later on to determine their timing of initiation, regardless of their subnuclear organization. This model is not mutually exclusive with the one proposed by the authors and should be discussed. This should be addressed in the paper.*

We have added a paragraph to the end of the Discussion addressing these papers (and one additional) in the context of our model, which we think fit quite well.

2) The view that the re-localization of Fkh1 origins does not depend on residual CDK activity in G1 is not fully supported by the experiment performed with clb5 clb6 mutants. Indeed, S-phase onset is only delayed by 30 minutes in this mutant. In principle, Clb3 and Clb4 could also support a residual CDK activity in alpha-factor arrested cells. It would have been more convincing to use a conditional cdc28 mutant.

We have done exactly as suggested using the *cdc28*-as allele to inactive all forms of Cdk1-based CDK activity. The results, (subsection “DDK- but not CDK-dependent step of replication initiation drives origin relocalization”, last paragraph and Figure 5D and new Figure 5—figure supplement 1) show that Cdc28 activity is not required for normal localization of ARS305 in G1 cells. We performed these analyses with cells synchronized with alpha-factor prior to Cdc28 inhibition (Figure 5D), and with G1 cells following inhibition of Cdc28 (no alpha-factor treatment) (Figure 5—figure supplement 1B, C). We have eliminated the prior CLB5/6 results for brevity.

3) A question that the authors do not address is why, in Figure 4—figure supplement 1 does Fkh1 bind non-Fkh1-activated origins? Or, put another way, why does Fkh1 not activate all the origins to which it binds? The manuscript focuses on Fkh1-regulated origins, but it would be important to show that the localization of late origins and Fkh1-independent origins does not depend on Fkh1 and/or DDK activity in G1.

In response to the first pair of questions: The definition of Fkh-act was statistically conservative and is based on origin differences in BrdU incorporation levels (WT vs. *fkh1∆fkh2∆)* in HU in our original study (Knott et al., 2012). We also observe that some later-firing origins are delayed in firing in *fkh1∆fkh2∆* cells in the absence of HU, including some telomere-proximal origins (see Figure 2A in Knott et al., 2012). Also, in subsequent studies we showed that overexpression of Fkh1 (Fkh1-OE) leads to increased early firing (in HU) of additional origins, and about half of the origins in previous Figure 4—figure supplement 3B that are designated “other origins” are Fkh1-OE-activated origins. So, most, if not all, Fkh-bound origins are stimulated, but not equally. For example, the number and spacing of Fkh1/2 sites appears to correlate with the level of stimulation (Looke et al., 2012). This is an interesting issue we are working on, but peripheral here, so to avoid this confusion, we have revised this figure (new Figure 4—figure supplement 1) to include only Fkh-activated origins, which directly corresponds to the statement in the text that this supplementary figure supports: “binding of Fkh1 to ARS305 and other Fkh-activated origins was largely unaffected by Cdc7 inhibition”

Regarding the second point: “it would be important to show that the localization […] does not depend on Fkh1 and/or DDK activity in G1,” we’d suggest that finding that late or Fkh1-independent origins’ localizations were due to Fkh1 and DDK would still be important, albeit less intuitive. Relocalization of specific genomic loci may have indirect consequences for positioning of sequences in *cis*. Nevertheless, we have analyzed ARS501 in WT and *fkh1∆* cells and observe that ARS501 is robustly peripheral in both strains. We have added this data in the text (subsection “Fkh1 globally regulates subnuclear positioning of early origins in G1 phase”, last paragraph) and new Figure 3—figure supplement 1C.

4) The authors' model is difficult to integrate into a more comprehensive model of sub-nuclear localization. In fkh1∆ fkh2∆, are all origins at the nuclear periphery? If so, is there any DNA in the nuclear interior? Why is ARS305^V-R^-∆ACS confined to the nuclear periphery? In authors' model is that origins bind to the periphery unless released by DDK, but in the case of ARS305^V-R^-∆ACS, it seems that the origin is required to drive the locus away from the periphery. Why would a non-origin locus bind to periphery? Do all non-origin loci bind to the periphery? These questions should be addressed by testing if the ARS305 ACS is required for FKH1-dependent localization at its endogenous locus.

In the absence of *FKH1*, most of the origins we tested are closer to the periphery though not necessarily at the periphery, and origin DNA is only a small fraction of total DNA, so we don’t think the interior is devoid of DNA. In the case of 305^V-R^-∆ACS, telomere proximity may maintain its peripheral localization (we suspect that Fkh1-induced relocalization of that locus likely results in release of the V-R telomere from the periphery as well, but this requires more work to address). To begin to address the additional questions posed by the reviewers, we have, as requested by the reviewers, analyzed 305∆ACS (and ∆BS) at the endogenous locus. The results are similar as with the V-R locus with both mutations resulting in peripheral localization. Deletion of the ACS causes even more peripheral distribution. This suggests two things: that the locus need not be tethered by an origin to be peripheral but that origin functionality promotes interior localization. Thus, the licensed origin lacking Fkh1/2 binding sites may still occasionally capture DDK and relocalize.

The idea that Rif is required for peripheral origin tethering is merely speculation; the finding that the locus remains peripheral in the absence of an origin does not contradict the model (there could be different reasons for the peripheral localization in the two cases). We have included a comment on this in the discussion of the model (Discussion, fifth paragraph). Regardless, we don’t think the limited analysis here is sufficient to answer these interesting questions raised by our study. Further work analyzing more loci, particularly more telomere-distal loci will be required to properly and fully answer these questions.